# Risk factors associated with high altitude sickness among travelers: A case control study in Himalaya district of Nepal

Sishir Poudel[1,2]*, Laxman Wagle[3], Mukhiya Ghale[2], Tara Prasad Aryal[4], Sushan Pokharel[1], Binay Adhikari[5]

1 BP Koirala Institute of Health Sciences, Dharan, Nepal, 2 Mustang Hospital, Mustang, Nepal, 3 Department of Internal Medicine, Ascension Saint Agnes Hospital, Baltimore, Maryland, United States of America, 4 Ministry of Health and Population, Kathmandu, Nepal, 5 Tuberculosis Treatment Center, Pokhara, Nepal

* sishir.poudel.sh@gmail.com

## Abstract

High elevation adventures are popular among travelers; however, they carry significant health risks, such as altitude sickness. This study aims to identify risk factors associated with high altitude sickness among travelers to Mustang district. A health-facility-based, age-sex matched 1:1 case-control study was conducted in Mustang district hospital, Nepal. Measurements included Acute Mountain Sickness/High Altitude Cerebral Edema/High Altitude Pulmonary Edema assessment via LLS questionnaire, demographics, medical history, ascent rate, and prophylactic medicine intake. Data were collected between September and November, 2023 via predesigned structured questionnaire by trained medical officers in, and analyzed using SPSS version 25. Using binary logistic regression, the study tested potential risk factors associated with altitude sickness. Ethical approval was obtained from the NHRC, and written informed consent was obtained from all participants. A total of 63 cases (individuals with altitude sickness) and 63 controls (without) were interviewed. The mean age of cases and controls was 48.5 years (SD = 16.5) and 48 years (SD = 16.9) respectively. 38 were rapid ascenders, and 88 were slow ascenders. Awareness of altitude sickness was reported by 65 individuals, with 36 taking prophylactic medication (Acetazolamide 125/250mg). Among cases, 8 experienced HACE, 42 had AMS, and 13 had HAPE. Rapid ascent (Adjusted Odds Ratio [AOR]: 6.41, 95% Confidence Interval [CI]: 2.36-17.54), individuals with a previous history of illness (AOR: 10.20, 95% CI: 2.70-38.46), and failing to take prophylactic medication (AOR: 10.01, 95% CI: 1.896-10.680) were linked to an increased risk of altitude sickness. Our study highlights the critical role of ascent speed, previous history of illness, and use of prophylactic measures in development of altitude sickness.

## Introduction

At high elevations, reduced barometric pressure causes hypobaric hypoxia, potentially leading to altitude illnesses such as Acute Mountain Sickness (AMS), High Altitude Cerebral

**Data availability statement:** Data is publicly available on figshare Link: https://dx.doi.org/10.6084/m9.figshare.27105400.

**Funding:** The author(s) received no specific funding for this work.

**Competing interests:** The authors have declared that no competing interests exist.

Edema (HACE), or High Altitude Pulmonary Edema (HAPE), ranging from mild discomfort to life-threatening emergencies, often manifesting within hours to days of ascent [1]. AMS, occurring at altitudes ≥2500 meters in unacclimatized individuals, presents nonspecific symptoms typically 4–12 hours post-ascent, peaking after the first night, but often resolves spontaneously with appropriate measures [2]. Symptoms of AMS, such as headache, nausea/vomiting, fatigue, dizziness, or difficulty sleeping, usually appear 6-12 hours after reaching high altitude, last 1-3 days, and, if untreated, can progress to HACE, potentially leading to coma or death within 24 hours [3]. HAPE, a non-cardiogenic pulmonary edema, can occur independently of AMS, typically developing 1-4 days after reaching altitudes above 2500 m [4].

Multiple factors such as home elevation, maximum altitude, ascent rate, age, gender, physical condition, exercise intensity, pre-acclimatization, genetics, and pre-existing conditions contribute to development of altitude sickness [5]. While men and women face similar AMS risks, previous AMS history significantly predicts future AMS development [5]. The influence of factors such as age, obesity, gender, fitness level, and substance use on the risk of altitude sickness among mountaineers remains uncertain [6]. Acclimatization level, ascent rate, and individual susceptibility are primary determinants [6].

Prophylactic acetazolamide, prescribed to prevent altitude sickness, typically ranges from 125 mg to 1 g per day, commonly administered as 125 mg or 250 mg twice daily [7]. However, it doesn't prevent symptoms if ascent is too rapid [7].

Globally, altitude-related illness incidence varies widely, with peaks around 4000m; for AMS, ranging from less than 10% to over 90%, and for HAPE/HACE, from less than 0.01% to 31% [8]. Highlighting the significant burden of altitude sickness, a hospital-based study in various trekking circuits in Nepal found an estimated altitude illness-related death incidence of 7.7 per 100,000 trekkers [9].

The Mustang district, known for Thorong La Pass (5416m) and Muktinath Temple (3800m), experiences heightened visitor flow due to improved roadways, reducing travel times [10]. However, rapid ascents from lower elevations raise concerns about altitude illness risk in the initial travel days [10]. Given the substantial impact of altitude sickness and the rising number of trekkers and tourists at high altitudes [10], it's crucial to identify risk factors and implement preventive measures to safeguard the safety and well-being of adventurers in such environments.

Therefore, this study aims to identify and analyze the risk factors associated with high altitude sickness among individuals visiting Mustang, providing valuable insights for prevention and management strategies in high-altitude destinations.

## Materials and methods

### Study design and setting

A health-facility-based, age-sex matched 1:1 case-control study was conducted from July 2023 to November 2023 in Mustang district of Gandaki Province, Nepal. Mustang is a mountainous district which lies at an elevation of 2,500- 3,000 meters above sea level. It is known for its several high-altitude attractions, including the world-renowned Thorong La Pass (5416m) in the Annapurna trekking circuit, the Muktinath Temple (3800m), Lo Manthang (3840m), and the Kora La Pass (4660m) which increase opportunities for trekkers. Mustang Hospital is the only available hospital providing basic, outpatient, and inpatients services to around 15000 residents in the district including the travelers. The routine health information in retrospect indicates the high presence of altitude sickness with peaks during the trekking season. This data also reflects the poor health system readiness towards management of AMS.

## Cases and controls selection

A participant was eligible to be included as a case, i.e., Altitude Sickness patient, if he/she, irrespective of place of residence, was a patient aged 18 years or older presenting in emergency services department with AMS/HACE/HAPE after his/her exposure to high altitude. A participant was eligible as a control if they were aged 18 years or older, presented to the emergency department without AMS/HACE/HAPE after high-altitude exposure, and had no history of altitude sickness. All participants resided below 2500m before exposure to high altitude, ensuring that acclimatization was not a confounding factor.

A hierarchical procedure was used to match the control with a case using the following variables and order (from the most to the least important): sex, same age ± 5 years.

## Measurements

The outcome of interest was the presence of AMS/HACE/HAPE. Subjects were evaluated using the LLS questionnaire, which rates the severity of four symptoms of AMS: headache, gastrointestinal symptoms (nausea/vomiting), fatigue/weakness, and dizziness/lightheadedness [11]. Each item is rated from 0 to 3, giving a maximum score of 12 [11]. As in the original description, subjects with LLQS ≥ 3 points in the presence of headache were diagnosed as AMS [11].

HACE will be diagnosed clinically if there is a change in mental status or ataxia in a person with AMS or a change in mental status and ataxia in a person without AMS [11]. For a patient to have HAPE, he or she must have at least two of the following symptoms: dyspnea at rest, cough, weakness or decreased exercise performance, chest tightness or congestion, and at least two of the following signs: crackling sounds or wheezing in at least one lung field, central cyanosis, tachypnea, and tachycardia [11].

The exposure variables included demographic characteristics, presence of medical co-morbidities, previous history of Altitude sickness, ascent rate, intake of prophylactic medicine, smoking and alcohol intake, awareness of (symptoms and vulnerable group) and symptoms related to high altitude sickness including travel history.

Slow ascent was defined as an elevation gain of no more than 600 meters per night above 3000m, reflecting a safe acclimatization approach for high altitudes [1].

## Sample size determination and recruitment

The number of participants for the study was calculated by using online based OpenEpi software, assuming the difference in proportion of AMS between two groups; AMS with rapid ascent was 58% (p1) while with slow ascent was 33% (p2) [12]. The minimum sample size was estimated to be 63 for cases and 63 for controls at 80% power, 95% confidence level, ratio of case and control as 1:1, expected odds ratio of ≥ 2, and assumption of exposure by control group. One control was selected randomly within the day of the selection of case. Controls were identified from same or different trekking or mountaineering groups as the cases.

## Data collection tools and study procedures

Patients were recruited over a 13 week period, from 01/09/2023 to 30/11/2023. After obtaining written informed consent, data was collected face to face using a structured, bilingual (English and Nepali) and pre-tested survey questionnaire prepared from existing related studies. Interviews were conducted in English or Nepali language, depending upon the language preference of the patient. To minimize recall bias, a Nepali calendar was used as a reference for recalling travel dates. Data was collected via KoboCollect app and exported as excel file before

analyzing in SPSS 25.0. All the participants were interviewed by the properly trained medical officers.

### Data analysis

Initially, the raw data was entered to Microsoft Excel, where thorough cleaning and consistency checks were performed. Descriptive statistics were presented as mean and standard deviation or frequency and percentage. Multivariable regression analysis was performed to measure the association between risk factors and Altitude Sickness. A p-value of less than 0.05 was considered statistically significant. All statistical analyses were performed using SPSS version 25 (IBM, Armonk, NY).

### Ethical considerations

Ethical approval was taken from Nepal Health Research Council. Permission was also taken from the hospital administration to approach participants for data collection (Ref No: 339/2023-308). Prior to data collection, the nature and purpose of the study was explicitly described verbally to participants and written informed consent from all the participants was obtained to begin data collection. Anonymity of participants was assured in the entirety of the study.

## Results

A total of 63 cases and 63 matched controls were interviewed for the study. Table 1 demonstrates the background characteristics of cases and controls.

The mean age for cases was 48.46 years (SD = 16.57), while controls had a mean age of 48 years (SD = 16.94). The overall mean age across both groups was 48.23 years (SD = 16.69). Majority of the participants were female (56%).

Most participants in the study were of Nepalese nationality (75%). The majority (65%) were classified as slow ascenders. A significant majority (65%), reported having no pre-existing health conditions. Most of participants had no history of altitude sickness (79%).

Only a total of 13 participants (10%) reported smoking during ascent, while none reported alcohol consumption during the same period. Furthermore, majority of individuals (79%) had no history of drug and medication use. Nearly half of the participants (48%) were unaware about altitude sickness. Prophylactic measures (Acetazolamide 125/250mg) were taken by only 36 (29%) of the total 126 individuals.

Among the cases, 67% had AMS, while 20% had HAPE, and 13% had HACE.

The results for the comparison of the health variables across cases and controls are shown in Table 1. There was significant difference between cases and controls with respect to ascent type, previous history of altitude sickness and intake of prophylactic medicine (p <0.01). All the remaining factors were not significantly associated (p ≥0.05) with altitude sickness.

The multivariate analysis in Table 3 revealed significant associations between various factors and the occurrence of altitude sickness. According to the final model, among ascent types, rapid ascenders demonstrated a substantially higher likelihood of altitude sickness compared to slow ascenders (AOR: 6.41, 95% CI: 2.36-17.54, p < 0.001). Similarly, individuals with a previous history of altitude sickness were significantly more predisposed to developing altitude sickness in the current study (AOR: 10.20, 95% CI: 2.70-38.46, p < 0.001). Moreover, individuals who didn't take prophylactic medicine (Acetazolamide 125/250mg) had a significantly higher risk of developing altitude sickness (AOR: 10.00, 95% CI: 2.70-33.33, p < 0.001).

Table 1. Background characteristics of cases and controls.

| Variables | Cases (N = 63) | | Controls (N = 63) | | p |
|---|---|---|---|---|---|
| | n | % | n | % | |
| **Ascent Type** | | | | | **0.000**[*] |
| Rapid | 29 | 46.03 | 9 | 14.29 | |
| Slow | 34 | 53.97 | 54 | 85.71 | |
| **Comorbidities** | | | | | 1.00 |
| Present | 19 | 30.16 | 19 | 30.16 | |
| Absent | 44 | 69.84 | 44 | 69.84 | |
| **Previous History** | | | | | **0.004**[*] |
| Yes | 19 | 30.16 | 6 | 9.52 | |
| No | 44 | 69.84 | 57 | 90.48 | |
| **Smoking** | | | | | 0.380 |
| Yes | 8 | 12.70 | 5 | 7.94 | |
| No | 55 | 87.30 | 58 | 92.06 | |
| **Drug History** | | | | | 1.00 |
| Present | 13 | 20.63 | 13 | 20.63 | |
| Absent | 50 | 79.37 | 50 | 79.37 | |
| **Awareness** | | | | | 0.050 |
| Yes | 27 | 42.86 | 38 | 60.32 | |
| No | 36 | 57.14 | 25 | 39.68 | |
| **Intake of prophylactic medicine(Acetazolamide 125/250mg)** | | | | | **0.000**[*] |
| Yes | 9 | 14.29 | 27 | 42.86 | |
| No | 54 | 85.71 | 36 | 57.14 | |

[*]Statistically significant at 0.05 level of significance.

As shown in Table 2, the association between awareness of altitude sickness and the intake of prophylactic medicine was found to be significant (p < 0.01).

Table 2. Association between awareness and intake of prophylactic medicine.

| AWARENESS | INTAKE OF PROPHYLACTIC MEDICINE | | P-value |
|---|---|---|---|
| | YES | NO | 0.000 |
| YES | 30(66.7%) | 35(97.2%) | |
| NO | 60(33.3%) | 1(2.8%) | |

## Discussion

The study demonstrates that rapid ascent, a previous history of altitude sickness, and lack of intake of prophylactic medicine were significant risk factors in the development of altitude sickness. Additionally, awareness of high altitude sickness was found to be associated with altitude sickness, although this association did not reach statistical significance.

Rapid ascenders were significantly more likely to experience altitude sickness compared to slow ascenders. This aligns with the literature, where studies by Kayser et al, Poudel KM et al, and Basynat B et al highlighted the significance of ascent rate in the development of altitude-related symptoms [13–15]. Slow ascent allows multiple acclimatization processes in the body to adapt to hypoxic environment thus decreasing incidence of altitude sickness. A study utilized ascent profile based model to predict altitude sickness in which a new metric called Accumulated Altitude Exposure (AAE) which was introduced to measure hypoxic dose, defined as

Table 3. Multivariable logistic regression analysis of risk factors associated with altitude sickness.

| Variables | Unadjusted | | Adjusted | |
|---|---|---|---|---|
| | OR (95% CI) | p-value | OR (95% CI) | p-value |
| **Ascent Type** | | **0.000***  | | **0.000*** |
| Rapid | 5.12(2.16-12.12) | | 6.41(2.36-17.54) | |
| Slow | Ref | | Ref | |
| **Comorbidities** | | 1.000 | | 0.873 |
| Present | 1.00(0.47-2.13) | | 0.89(0.24-3.45) | |
| Absent | Ref | | Ref | |
| **Previous History** | | **0.004*** | | **0.001*** |
| Yes | 4.10(1.51-11.11) | | 10.20(2.70-38.46) | |
| No | Ref | | Ref | |
| **Smoking** | | 0.380 | | 0.572 |
| Yes | 1.69(0.52-5.56) | | 0.65(0.15-2.86) | |
| No | Ref | | Ref | |
| **Drug History** | | 1.000 | | 0.791 |
| Yes | 1.00(0.42-2.38) | | 0.81(0.18-3.7) | |
| No | Ref | | Ref | |
| **Awareness** | | 0.050 | | 0.460 |
| Yes | Ref | | Ref | |
| No | 2.03(1.00-4.12) | | 1.47(0.53-4.03) | |
| **Intake of prophylactic medicine(Acetazolamide 125/250mg)** | | **0.000*** | | **0.001*** |
| Yes | Ref | | Ref | |
| No | 4.55(1.89-11.11) | | 10.00(2.70-33.33) | |

*Statistically significant at 0.05 level of significance.

OR = Odds Ration, CI = Confidence interval

the product of altitude elevation (km) and the number of days (d) spent at that altitude before ascending to 4 km [16]. It showed that for each 1 km-d increase in AAE, the likelihood of falling ill decreases by 41.3%, thus highlighting role of ascent rate in altitude sickness [16].

The presence of a previous history of altitude sickness emerged as a significant predictor of susceptibility to altitude sickness in our study. Individuals with no prior history were markedly less likely to develop altitude sickness. This finding is consistent with the work of Schneider et al, who reported a substantial increase in AMS prevalence with higher scores of a history of AMS (OR 2.9, 95% CI, 2.1– 4.1) [12]. However in the study by K Mairer et al history of AMS did not influence the prevalence of AMS [17].

Awareness of altitude sickness exhibited a statistically non-significant association with its occurrence. This suggests that while awareness may influence the likelihood of altitude sickness, other unexplored factors may also contribute to its development. The literature findings from a study conducted on trekkers in the Nepalese Himalaya reported a 63% prevalence of AMS despite 80% of trekkers having elementary knowledge of AMS [13]. A 12-year comparative study conducted on trekkers in the Nepalese Himalaya revealed significant changes in altitude-related health awareness and prevalence of AMS [18]. The awareness of AMS increased from 80% to 95%, while the prevalence of AMS decreased from 43% to 29% [18]. Awareness of altitude-related health hazards was shown to influence other risk factors like ascent rates, medication usage leading to a reduced prevalence of AMS [18].

The protective effect of prophylactic medicine (acetazolamide 125/250mg) against altitude sickness was significant in our study. Individuals who took prophylactic medicine

had a substantially lower risk of developing altitude sickness. These findings align with the meta-analysis conducted by Ried et al, which demonstrated the efficacy of acetazolamide in preventing AMS [7]. Theoretically, medications aimed at preventing AMS (acetazolamide) may also reduce the risk of HACE/HAPE; however, this hypothesis has not been thoroughly investigated through systematic research [1].

Our study provides valuable insights into the risk factors associated with altitude sickness, highlighting the importance of ascent type, previous history, awareness, and prophylactic measures. This study offers a fresh perspective on the risk factors for high-altitude sickness by being among the limited case-control studies available. While earlier research has highlighted major risk factors, this study is one of the few to investigate how awareness and prior sickness experiences influence preventive measures. However, it is essential to acknowledge certain limitations, including small sample size and the potential for recall bias in self-reported data and the influence of unmeasured confounders like physical fitness, genetic predisposition, and other factors.

## Conclusion

In conclusion, the results of our study emphasize the need for careful planning of ascent rates and the importance of preventive measures for travelers. These insights have practical implications for individuals ascending to high altitudes, providing evidence-based guidance for the prevention of altitude-related illnesses. Further research with a larger sample size and a prospective study design is warranted to explore additional factors contributing to altitude sickness and to refine preventive strategies for individuals at risk.

## Supporting information

**S1 Checklist. Inclusivity in global research.**
(DOCX)

## Acknowledgement

We express our gratitude to NHRC and Mustang District Hospital staffs for their support. We appreciate the time contributed by participants for this important study.

## Author contributions

**Conceptualization:** Sishir Poudel.

**Data curation:** Sishir Poudel, Laxman Wagle, Mukhiya Ghale, Binay Adhikari.

**Formal analysis:** Sishir Poudel, Laxman Wagle, Tara Prasad Aryal, Sushan Pokharel.

**Investigation:** Sishir Poudel, Mukhiya Ghale, Binay Adhikari.

**Methodology:** Sishir Poudel, Laxman Wagle, Mukhiya Ghale, Tara Prasad Aryal, Sushan Pokharel.

**Project administration:** Laxman Wagle.

**Resources:** Sishir Poudel.

**Software:** Sishir Poudel.

**Supervision:** Sishir Poudel, Laxman Wagle, Mukhiya Ghale, Tara Prasad Aryal, Sushan Pokharel.

**Validation:** Sishir Poudel, Mukhiya Ghale.

**Visualization:** Sishir Poudel, Laxman Wagle, Tara Prasad Aryal.

**Writing – original draft:** Sishir Poudel.

**Writing – review & editing:** Sishir Poudel, Laxman Wagle, Mukhiya Ghale, Tara Prasad Aryal, Sushan Pokharel, Binay Adhikari.

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
