## [Decision Letter · Decision Letter 0]

9 Sep 2024

PGPH-D-24-00773

Risk factors associated with high altitude sickness among travelers: A case control study in Himalaya district of Nepal

Dear Dr. Poudel,

Thank you for submitting your manuscript to PLOS Global Public Health. After careful consideration, we feel that it has merit but does not fully meet PLOS Global Public Health’s publication criteria as it currently stands. Therefore, we invite you to submit a revised version of the manuscript that addresses the points raised during the review process.

Please note that we have only been able to secure a single reviewer to assess your manuscript. We are issuing a decision on your manuscript at this point to prevent further delays in the evaluation of your manuscript. Please be aware that the editor who handles your revised manuscript might find it necessary to invite additional reviewers to assess this work once the revised manuscript is submitted. However, we will aim to proceed on the basis of this single review if possible. 

We look forward to receiving your revised manuscript.

Kind regards,

Joanna Tindall, PhD

Staff Editor

Journal Requirements:

Additional Editor Comments (if provided):

Reviewers' comments:

Reviewer's Responses to Questions

**Comments to the Author**

1. Does this manuscript meet PLOS Global Public Health’s publication criteria ? Is the manuscript technically sound, and do the data support the conclusions? The manuscript must describe methodologically and ethically rigorous research with conclusions that are appropriately drawn based on the data presented.

Reviewer #1: Yes

2. Has the statistical analysis been performed appropriately and rigorously?

Reviewer #1: Yes

3. Have the authors made all data underlying the findings in their manuscript fully available (please refer to the Data Availability Statement at the start of the manuscript PDF file)?

Reviewer #1: Yes

4. Is the manuscript presented in an intelligible fashion and written in standard English?

Reviewer #1: Yes

5. Review Comments to the Author

Reviewer #1: This case-control study assesses the risk factors associated with altitude illness and highlights the importance of various factors in relation to the development of altitude illness.

INTRODUCTION

Line 61 – Need to define HACE and HAPE before using abbreviations.

Lines 107-108 the sentence ‘However, rapidly ascending from lower

107 elevations to such hypobaric hypoxia environment, thereby increasing the risk for developing high altitude illness for many travelers’ – not sure this is an appropriate statement for the methods section. You already state/imply this in the introduction in lines 84-86 anyways.

Line 115: ‘Irrespective of residence altitude’ – this implies that they could be acclimatized? Or high altitude resident? And in relation to this, what if participants were matched that resided at different altitudes?

Lines 125: LLS is a symptom severity questionnaire that is highly subjective rather than a clinical evaluation. Be careful with the wording here.

Lines 126 – 128: Sleep has been removed from the most current LLS scoring criteria (2018). In this instance you must specifiy reference the use of the old version of the questionnaire which includes sleep.

Lines 130: Again you’ve referenced the newest version of the LLS, however, this particular version has removed sleep from the scoring system.

Line 155: ‘standard questionnaire’ – is this a healthy history questionnaire? what information was this gathering? General demographics?

Line 162 – Consistency checks? Were these checked by 2 reviewers? Or were and certain percentage of the data randomly re-checked with the hard data sheets?

Ethical considerations – perhaps this is a journal specific thing but typically this goes at the beginning of the methods section. Does the NHRC provide an ethical review number, worth referencing this specifically.

It is somewhat unclear how ‘awareness’ was assessed and what exactly ‘awareness’ was for? Awareness that high-altitude illnesses exist, and knowing what the symptoms are? Or was it awareness of medications used for altitude illness? Or both. Authors should be more clear about the assessment of ‘awareness’ in the methods section.

RESULTS

Lines 212-214: Unsure why worth pointing out an insignificant association especially given it is already presented in the Table 3. Avoid duplicating results in table and text.

Also unclear to reader why there are adjusted and unadjusted OR presented? Is this for if they were calculated as a singular variable and then adjusted for multivariate analysis?

DISCUSSION

Line 261- You already define and use the abbreviation for AMS so you need not to do this again in the discussion.

Line 269 – acetazolamide in parenthesis should be changed to (i.e., acetazolamide). I also believe through the paper that acetazolamide is capitalised , however, as a generic drug, generally, this is not capitalized where as, Diamox (the brand name) would be capitalised.

General thought for future directions - Were individuals with history of altitude sickness also more likely to take medication to prevent sickness? And were those with previous sickness also the individuals who had the most ‘awareness’? Perhaps this is beyond the scope but it would be interesting to know whether ‘awareness’ is essentially spawned from people getting sick or whether they are actually being educated about altitude sickness. Perhaps an important thing to know moving forward in terms of how to improve the amount of people getting sick!

The most important thing to highlight is how is this study contributes anything different to existing studies in regards to further understanding of risk factors related to sickness. After all you are able to state the primary and significant risk factors in the introduction already, so it will me important to highlight how this study contributes something additional/provide something different to the existing literature. For example, is it the only (or one of very limited) case-control study of it’s kind? Etc.

6. PLOS authors have the option to publish the peer review history of their article (what does this mean? ). If published, this will include your full peer review and any attached files.

**Do you want your identity to be public for this peer review?** For information about this choice, including consent withdrawal, please see our Privacy Policy .

Reviewer #1: No

---

## [Decision Letter · Decision Letter 1]

11 Nov 2024

PGPH-D-24-00773R1

Risk factors associated with high altitude sickness among travelers: A case control study in Himalaya district of Nepal

Dear Dr. Poudel,

Thank you for submitting your manuscript to PLOS Global Public Health. After careful consideration, we feel that it has merit but does not fully meet PLOS Global Public Health’s publication criteria as it currently stands. Therefore, we invite you to submit a revised version of the manuscript that addresses the points raised during the review process.

The manuscript has been evaluated by two reviewers, and their comments are available below.

The reviewers have raised a number of concerns. They request improvements to the reporting of methodological aspects of the study, and to the discussion of the study limitations.

Could you please carefully revise the manuscript to address all comments raised?

We look forward to receiving your revised manuscript.

Kind regards,

Helen Howard

Staff Editor

Additional Editor Comments (if provided):

Reviewers' comments:

Reviewer's Responses to Questions

**Comments to the Author**

1. If the authors have adequately addressed your comments raised in a previous round of review and you feel that this manuscript is now acceptable for publication, you may indicate that here to bypass the “Comments to the Author” section, enter your conflict of interest statement in the “Confidential to Editor” section, and submit your "Accept" recommendation.

Reviewer #1: All comments have been addressed

Reviewer #2: (No Response)

2. Does this manuscript meet PLOS Global Public Health’s publication criteria ? Is the manuscript technically sound, and do the data support the conclusions? The manuscript must describe methodologically and ethically rigorous research with conclusions that are appropriately drawn based on the data presented.

Reviewer #1: Yes

Reviewer #2: Yes

3. Has the statistical analysis been performed appropriately and rigorously?

Reviewer #1: Yes

Reviewer #2: Yes

4. Have the authors made all data underlying the findings in their manuscript fully available (please refer to the Data Availability Statement at the start of the manuscript PDF file)?

Reviewer #1: Yes

Reviewer #2: Yes

5. Is the manuscript presented in an intelligible fashion and written in standard English?

Reviewer #1: Yes

Reviewer #2: Yes

6. Review Comments to the Author

Reviewer #1: Authors have adequately addressed all previous comments and concerns. I have no further comments.

Reviewer #2: Reviewer Comments (better formatted comments attached as a supplement)

The manuscript addresses a highly relevant topic concerning the risk factors associated with high-altitude sickness (AMS, HAPE, HACE) among travelers to the Mustang district of Nepal. The growing popularity of adventure tourism and trekking in high-altitude regions has made this topic particularly timely and significant. The authors have approached this problem with a well-defined case-control study, which is an appropriate design for exploring associations between exposure factors and altitude sickness. Additionally, the emphasis on a specific region, Mustang, known for its significant trekking activity, adds valuable context and geographical specificity to the findings, increasing their relevance for both public health practitioners and tourism policymakers.

The manuscript's major strengths include its clear research question, the use of a well-matched case-control design (1:1 age-sex matching), and the rigorous use of logistic regression to determine associations between multiple potential risk factors and the development of altitude sickness. The statistical analysis is performed appropriately, and the authors provide a detailed discussion of the results, incorporating evidence from previous studies to contextualize their findings.

The discussion highlights the importance of ascent type, previous altitude sickness history, and prophylactic medication intake (acetazolamide) as significant risk factors for developing altitude sickness. These findings contribute to the broader understanding of risk factors that can be managed or mitigated through preventive measures, making the manuscript of potential utility for travel planners, public health experts, and individual adventurers.

While the manuscript presents several strong points, it also contains some limitations that need to be addressed for it to achieve its full potential. Firstly, the sample size is relatively small (63 cases and 63 controls). Although the case-control design helps address power limitations, the small sample size may affect the generalizability of the findings. Future studies with a larger cohort would further validate the associations observed here.

The authors used the Lake Louise Score (LLS) for diagnosing AMS, HAPE, and HACE. However, the LLS is a subjective assessment tool, and its use as the sole method for clinical evaluation should be carefully framed to avoid overstating its diagnostic power. The term "clinically evaluated" should be revised, as LLS is a symptom score rather than a diagnostic method. Moreover, some important confounders, such as physical fitness level and genetic predisposition, were not accounted for, which could limit the interpretation of the results.

Another issue lies in the presentation of some results and methodological descriptions. Certain sections, such as the criteria for control selection and sample size calculation, require more clarity. Additionally, there are redundant statements in the methods and introduction sections, which should be streamlined for a more concise presentation. The suggestions provided in the table below offer specific guidance for improving these aspects.

Detailed suggested edits:

Original Text Suggested Edits

Abstract: Line 21: "High elevation adventure is a traveler’s choice. However, it has many health repercussions like altitude sickness." "High elevation adventures are popular among travelers; however, they carry significant health risks, such as altitude sickness."

Introduction: Line 61: "AMS, HACE, or HAPE, ranging from mild discomfort to life-threatening emergencies" "Acute Mountain Sickness (AMS), High Altitude Cerebral Edema (HACE), or High-Altitude Pulmonary Edema (HAPE), ranging from mild discomfort to life-threatening emergencies."

Introduction: Line 67: "AMS symptoms like headache, nausea/vomiting, fatigue, dizziness, or difficulty sleeping usually appear 6-12 hours post high-altitude arrival, lasting 1-3 days; untreated AMS can lead to HACE, with altered mental status and potential coma or death within 24 hours." "Symptoms of AMS, such as headache, nausea/vomiting, fatigue, dizziness, or difficulty sleeping, usually appear 6-12 hours after reaching high altitude and last 1-3 days. If untreated, AMS can progress to HACE, potentially leading to coma or death within 24 hours."

Introduction: Line 73: "Factors like age, obesity, gender, fitness level, and substance use in altitude sickness risk among mountaineers remain uncertain." "The influence of factors such as age, obesity, gender, fitness level, and substance use on the risk of altitude sickness among mountaineers remains uncertain."

Methods: Lines 107-108: "However, rapidly ascending from lower elevations to such hypobaric hypoxia environment, thereby increasing the risk for developing high-altitude illness for many travelers." Remove this statement, as it repeats information already provided in the introduction.

Methods: Line 115: "Irrespective of residence altitude" Clarify that participants were not acclimatized individuals. Suggestion: "All participants resided below 2500m before exposure to high altitude, ensuring that acclimatization was not a confounding factor."

Methods: Line 119: "A participant was eligible to be included as a control i.e. non-Altitude Sickness patient, if he/she, irrespective of place of residence, is patients aged 18 years or older presenting in emergency services department without AMS/HACE/HAPE after his/her exposure to high altitude." "A participant was eligible as a control if they were aged 18 years or older, presented to the emergency department without AMS/HACE/HAPE after high-altitude exposure, and had no history of altitude sickness."

Methods: Line 125: "Subjects were clinically evaluated for AMS using the LLS, a questionnaire that rates the severity of 4-5 symptoms of AMS." Replace "clinically evaluated" with "evaluated using the LLS questionnaire," as the LLS is subjective rather than a clinical diagnosis.

Methods: Line 140: "Slow ascent was defined as an elevation gain of no more than 600 meters per night (above 3000m), reflecting approach for safe acclimatization to altitudes exceeding 3000 meters." "Slow ascent was defined as an elevation gain of no more than 600 meters per night above 3000m, reflecting a safe acclimatization approach for high altitudes."

Results: Line 182: "The study group consisted most of individuals of Nepalese nationality (75%). Majority of participants were classified as slow ascenders (65%)." "Most participants in the study were of Nepalese nationality (75%). The majority (65%) were classified as slow ascenders."

Results: Line 195: "Among the cases, majority (67%) had Acute Mountain Sickness (AMS) while others presented with High Altitude Pulmonary Edema (HAPE)(20%) and High Altitude Cerebral Edema (HACE)(13%)." "Among the cases, 67% had Acute Mountain Sickness (AMS), while 20% had High Altitude Pulmonary Edema (HAPE), and 13% had High Altitude Cerebral Edema (HACE)."

Discussion: Line 233: "Rapid ascenders demonstrated a significantly higher likelihood of experiencing altitude sickness compared to their slow ascending counterparts." "Rapid ascenders were significantly more likely to experience altitude sickness compared to slow ascenders."

Conclusion: Line 282: "Our case-control study contributes to the understanding of risk factors for altitude sickness, emphasizing the significance of ascent speed, previous history, and prophylactic measures." Consider highlighting the implications for travel planning and tourism management: "These findings emphasize the need for careful planning of ascent rates and the importance of preventive measures for travelers."

In conclusion, the manuscript presents a valuable investigation into the risk factors associated with altitude sickness among travelers in Mustang, Nepal. The use of a case-control study design and logistic regression analysis provides a solid foundation for identifying important risk factors, such as ascent speed, previous history of altitude sickness, and use of prophylactic medication. However, limitations regarding sample size, subjective diagnostic tools, and some confounders need to be addressed to strengthen the study's reliability and generalizability. I recommend minor revisions to address these concerns, improve clarity, and add supporting references where necessary. With these adjustments, the manuscript would be suitable for publication in its current journal.

7. PLOS authors have the option to publish the peer review history of their article (what does this mean? ). If published, this will include your full peer review and any attached files.

**Do you want your identity to be public for this peer review?** For information about this choice, including consent withdrawal, please see our Privacy Policy .

Reviewer #1: No

Reviewer #2: **Yes: ** Wanjun Gu

---

## [Decision Letter · Decision Letter 2]

14 Jan 2025

Risk factors associated with high altitude sickness among travelers: A case control study in Himalaya district of Nepal

PGPH-D-24-00773R2

Dear Dr. Poudel,

We are pleased to inform you that your manuscript 'Risk factors associated with high altitude sickness among travelers: A case control study in Himalaya district of Nepal' has been provisionally accepted for publication in PLOS Global Public Health.

Best regards,

Julia Robinson

Executive Editor

Reviewer Comments (if any, and for reference):

Reviewer's Responses to Questions

**Comments to the Author**

1. If the authors have adequately addressed your comments raised in a previous round of review and you feel that this manuscript is now acceptable for publication, you may indicate that here to bypass the “Comments to the Author” section, enter your conflict of interest statement in the “Confidential to Editor” section, and submit your "Accept" recommendation.

Reviewer #2: All comments have been addressed

2. Does this manuscript meet PLOS Global Public Health’s publication criteria ? Is the manuscript technically sound, and do the data support the conclusions? The manuscript must describe methodologically and ethically rigorous research with conclusions that are appropriately drawn based on the data presented.

Reviewer #2: Yes

3. Has the statistical analysis been performed appropriately and rigorously?

Reviewer #2: Yes

4. Have the authors made all data underlying the findings in their manuscript fully available (please refer to the Data Availability Statement at the start of the manuscript PDF file)?

Reviewer #2: Yes

5. Is the manuscript presented in an intelligible fashion and written in standard English?

Reviewer #2: Yes

6. Review Comments to the Author

Reviewer #2: The reviewer believes that the authors have nicely addressed all the comments and provided detail oriented feedback on the questions that the reviewer has regarding their manuscript. The reviewer would thus like to endorse the manuscript for publication.

7. PLOS authors have the option to publish the peer review history of their article (what does this mean? ). If published, this will include your full peer review and any attached files.

**Do you want your identity to be public for this peer review?** For information about this choice, including consent withdrawal, please see our Privacy Policy .

Reviewer #2: **Yes: ** Wanjun Gu
